# MAXIMUM CATEGORICAL CROSS ENTROPY (MCCE): A NOISE-ROBUST ALTERNATIVE LOSS FUNCTION TO MITIGATE RACIAL BIAS IN CONVOLUTIONAL NEURAL NETWORKS (CNNS) BY REDUCING OVERFITTING

## ABSTRACT

Categorical Cross Entropy (CCE) is the most commonly used loss function in deep neural networks such as Convolutional Neural Networks (CNNs) for multi-class classification problems. In spite of the fact that CCE is highly susceptible to noise; CNN models trained without accounting for the unique noise characteristics of the input data, or noise introduced during model training, invariably suffer from overfitting affecting model generalizability. The lack of generalizability becomes especially apparent in the context of ethnicity/racial image classification problems encountered in the domain of computer vision. One such problem is the unintended discriminatory racial bias that CNN models trained using CCE fail to adequately address. In other words, CNN models trained using CCE offer a skewed representation of classification performance favoring lighter skin tones.

In this paper, we propose and empirically validate a novel noise-robust extension to the existing CCE loss function called Maximum Categorical Cross-Entropy (MCCE), which utilizes CCE loss and a novel reconstruction loss, calculated using the Maximum Entropy (ME) measures of the convolutional kernel weights and input training dataset. We compare the use of MCCE with CCE-trained models on two benchmarking datasets, colorFERET and UTKFace, using a Residual Network (ResNet) CNN architecture. MCCE-trained models reduce overfitting by 5.85% and 4.3% on colorFERET and UTKFace datasets respectively. In cross-validation testing, MCCE-trained models outperform CCE-trained models by 8.8% and 25.16% on the colorFERET and UTKFace datasets respectively. MCCE addresses and mitigates the persistent problem of inadvertent racial bias for facial recognition problems in the domain of computer vision.

## 1 INTRODUCTION

Convolutional Neural Networks (CNNs) offer state-of-the-art results in computer vision tasks He et al. (2016); Szegedy et al. (2015); Simonyan & Zisserman (2014) but are susceptible to inherent noises in the input training data preempting overfitting on the input data during information propagation. When new data is presented, overfit models do not generalize well and offer significantly lower classification performance, exacerbating the problem of bias towards a specific subset of data. The fundamental learning theory behind CNNs is to approximate an underlying $d$-dimensional interpolated function $f(\mathbf{X}) \in \mathbb{R}^d$ by using information from $n$ number of $d$-dimensional input vectors $\mathbf{X} = \{\mathbf{x}_1, \mathbf{x}_2, \cdots, \mathbf{x}_n\}$ where $\mathbf{x}_i = <x^1, x^2, \cdots, x^d>$ and $i, d \in \mathbb{Z}_{>0}$ Maiorov (2006). The problem of approximation is theoretically non-linear and there is empirical evidence to support the assertion that CNNs simply memorize the input training data Zhang et al. (2016).

Overfitting occurs when the internal parameters of a CNN model are finely tuned to the unique variances of the input training data that it perfectly models its characteristics Hawkins (2004). Misclassification occurs when overfit models are unable to distinguish between overlapping variances for different classes of images. Reducing overfitting is also difficult since establishing a theoretical understanding or analyzing the mechanisms of learning in CNNs for non-convex optimization problems such as image classification is generally not well understood Shamir (2018).

A simple way to reduce overfitting is to train models using a very large number of images Shorten & Khoshgoftaar (2019), such as the ImageNet dataset consisting of millions of training images used for the purpose of natural image classification. While using big data solutions might mask the underlying problem of model overfitting, acquisition of clean/noise-free labeled data for supervised model training is challenging. The problem of data acquisition is compounded further by ethical, societal, and practical concerns when dealing with facial datasets, especially for the task of race or gender classification.

Another key challenge while creating datasets is the consideration that needs to be made on the distribution of data amongst the multiple classes along with the variability of data within an individual class. Unbalanced datasets where the data distribution of images is not equal for all the classes introduces bias during model training Ganganwar (2012). The only viable solution to rectify imbalanced datasets is to augment or supplement datasets with new images which as mentioned before is an ongoing challenge. To the best of our knowledge, there is no research/work undertaken to optimize data distribution of the convolutional kernel weights during model training. We hypothesize that balancing convolutional kernel data, during model training could aide in mitigating bias and increase classification performance through alleviating the severity of inherent noise.

Some researchers attribute racial bias of CNN models to noises in the training data and associated labels proposing alternate loss functions like Mean Absolute Error (MAE) Ghosh et al. (2017) to commonly used loss functions like *Categorical Cross Entropy (CCE)*, as explained in Section 2.1. MAE was proposed as a noise-robust alternative to mitigate the susceptibility of CNNs to noise, but as Zhang & Sabuncu (2018) asserts, MAE is not applicable for complex natural image datasets like ImageNet and as such it is not considered in this paper. The task of classifying race in human faces is established to be more complex than natural image classification because there exists a narrow range of possible variations in features between human faces of different races, especially when skin tone is not the major determining factor for racial identity Fu et al. (2014).

In this paper, we explore the problem of overfitting with respect to racial classification by assessing the train-test divergence to quantify the degree of generalizability where a higher train-test divergence indicates a greater degree of model overfitting on the training data. We also propose a novel extension to the commonly used CCE loss function using Maximum Entropy (ME) Hartley (1928) measures, called Maximum Categorical Cross Entropy (MCCE). MCCE loss calculations are determined by taking into account the distribution of convolutional kernel weights during model training and the traditional CCE loss. Most related works explore model over-parameterization Zhang et al. (2019) or under-parameterization Soltanolkotabi et al. (2018) with unrealistic assumptions made about the distribution of input data; we do not make any such assumptions.

The contributions of this paper are as follows:

- We propose a novel extension to the Categorical Cross Entropy (CCE) loss function using Maximum Entropy (ME) measures known as Maximum Categorical Cross Entropy (MCCE) loss to reduce model overfitting.

- We empirically validate the MCCE loss function with respect to model overfitting using train-test divergence as a metric and evaluate generalizability across datasets by using cross-validation testing.

## 2 BACKGROUND

Section 2.1 presents an understanding of how CCE loss is calculated. Sections 2.2 and **??** detail how kernel regularization and batch normalization influence CCE loss with their limitations. Section 2.3 provides the theoretical background of Maximum Entropy (ME) and methods to calculate ME along with estimating the reconstruction loss.

### 2.1 CATEGORICAL CROSS-ENTROPY (CCE) LOSS

The most commonly used loss function is the Categorical Cross-Entropy (CCE) loss given in Equation (1), which is a measure of difference between the probability distributions of one-hot encoded CNN computed class labels and ground truths. CNN classification uses a softmax function to calculate the required probability distributions Goodfellow et al. (2016).

$$H(p,q) = \Sigma_{i=1}^{n} = p(x_i) \; log \; q(x_i) \quad Where, \mathbf{x}_i \in \mathbf{X} \tag{1}$$

In Equation (1), $q(x_i)$ and $p(x_i)$ represent the probability distributions of the one-hot encoded CNN predicted class labels and ground truths respectively for an input data vector $\mathbf{x}_i$. Given that CNN model training introduces noises during convolutional operations or information propagation and that any inherent noise present in the input data can significantly affect model performance, a noise-robust alternative to CCE would help improve classification performance and mitigate bias. This is the reason why stochastic optimizers and gradient descent algorithms function using the framework of maximum likelihood estimation.

### 2.2 KERNEL REGULARIZATION

The intuition behind regularization is that of Ockham's razor to penalize complex models and to promote simpler models during training. Unlike empirical risk minimization which only considers loss minimization, regularization was proposed to minimize structural risk which considers both complexity and loss minimization. The most prominent and simple kernels that greatly minimize loss are selected Bilgic et al. (2014). Model complexity is represented in two ways, as a function of the total number of features with non-zero weights ($L_1$) or as a function of all the weights of all the features in a model ($L_2$). $L_2$ regularization is most commonly used in computer vision tasks for CNN models such as ResNet. Model complexity can be quantified using the $L_2$ regularization formula given in Equation (2), defined by using the sum of squares of all the feature weights as the regularization term Cortes et al. (2012).

$$||\omega||^2 = \omega_1^2 + \omega_2^2 + \omega_3^2 + \omega_n^2 \tag{2}$$

In Equation (2), the magnitude of the absolute value of the weight $\omega$ indicates complexity. Feature weights close to zero have no significant impact on model complexity, while large outlier weight values have a more pronounced impact on $\omega$. The quantity of feature weights $n$ determined using the number of trainable model parameters also contribute greatly to $\omega$ and model complexity. Furthermore, kernel regularization as it is implemented currently for CCE loss utilizes CNN computed label errors and does not take the data distribution of the convolutional kernels into account.

### 2.3 MAXIMUM ENTROPY AND RECONSTRUCTION LOSS

The use of Maximum Entropy (ME) for applications such as convolutional kernel analysis is justified since ME is the only consistent way of selecting a single discrete data point from the set of input data vectors to best fit the regression curve, proven axiomatically in Shore & Johnson (1980); Johnson & Shore (1983).

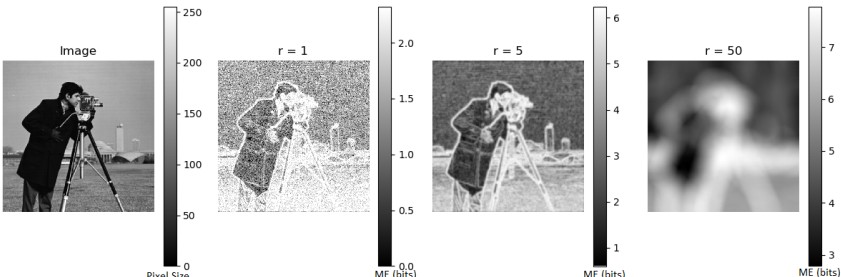

Figure 1: Difference in ME measurement for changes in the radius $r$ used to measure the complexity contained in a local neighborhood.

A method to approximate ME for digital images is through the use of distributed normalized histograms Gonzalez & Woods (2007); Jain (1989). The open-source SciKit-image processing library written in Python can be used to calculate the ME measures for images Virtanen et al. (2020).

Entropy in images is related to the complexity contained in a given neighborhood, computed by using a circular disk with a radius of $r$. The disk is used to measure minute variations in local grayscale level distribution. The maximum entropy for an image depends on the number of gray levels, an 8 bit image has 256 gray levels (0-255) which has a theoretical maximum entropy of $log_2(2^8) = 8$ bits per pixel. Changing the value of $r$ can invariable produce higher or lower ME measure as illustrated in Figure 1. Similarly higher or lower ME values will be obtained while measuring convolutional kernel weights. A decrease in ME divergence can be observed in Figure 1 for $r$ values of 5 and 50 relative to $r$ values of 1 and 5. A significant difference in spatial/semantic information in the images can be seen with greater $r$ values, which suggests loss in precision during approximation.

ME measures for color images require the computation on each of the three color channels, Red (R), Green (G) and Blue (B) i.e. RGB separately and averaging the result. The averaged ME measures for images in the *colorFERET* and *UTKFace* datasets are *2.09* and *2.25* bits per pixel respectively using an $r$ value of 1. The amount of time taken to calculate the ME measures is insignificant as the ME calculation script can be executed in parallel on the CPU, while CNN model training occurs on the GPU, as *evidenced in the supplementary data uploaded*. Solutions other than ME for image reproduction/reconstruction from noisy or incomplete measurements such as, the use of non-linear variations on fourier transformations fail when convolutional kernels are incorporated Donoho et al. (1990). Furthermore, ME reconstruction has been shown to provide superior noise suppression while mostly preserving de-emphasized structural noise near the baseline (relative to high signal information) Donoho et al. (1990).

Accurate reconstructions can be approximated using a 1D projection of any underlying function which is reduced to $g(\mathbf{X}) \in \mathbb{R}^d$ such that $\mathbf{x}_i \in \mathbf{X}$ Reis & Roberty (1992). As discussed in Section 1, the underlying functional representation of the input dataset is $f(\mathbf{X})$, the difference between the true representation $f(\mathbf{X})$ and the ME reconstruction approximation $g(\mathbf{X})$ is the reconstruction loss for the input dataset. Results presented in Reis & Roberty (1992), indicate that reconstructions using accurate and noisy data had insignificantly small variations compared to the original, attesting to the noise-robust ability of using ME measures for reconstruction. This noise averse characteristic of ME is especially important for race classification as lighting or ISO parameters of the input images can significantly affect the performance of CNN models. Reconstruction loss is described as the convolutional kernel data loss whereas CCE can be characterized as a class label loss.

## 3 MAXIMUM CATEGORICAL CROSS-ENTROPY (MCCE)

The classification of data in CNNs primarily depends on the convolutional kernels represented by their weights. Optimization of kernel weights using a loss function is performed to ensure a closer approximation to the underlying function $f(\mathbf{X})$ is achieved. As discussed in Section 2.1, CCE is a measure of difference between two probability distributions, the ground truth and CNN computed label for a class $C$. The drawback of CCE is that it only considers class label errors and does not account for the distribution of the convolutional kernel weights. The estimation of kernel weight probability distributions is critical in knowing the state of model training and learning capacities which could enhance classification performance, and MCCE is proposed to rectify this limitation.

The Maximum Categorical Cross-Entropy (MCCE) loss function monitors the data distribution of convolutional kernel weights using ME measures along with traditional CCE loss and penalizes models which are overly complex. Apriori knowledge of the entropic distribution of the input data can be computed using ME measures which is used as a baseline to monitor convolutional kernel weight distributions and penalize models with greater divergences. It is well understood that maximizing entropy measures using even partial information (such as from convolutional kernel weights) can enhance the estimation of probability distributions Macqueen & Marschak (1975) used extensively to calculate CCE loss.

The main criterion for producing high quality reconstruction approximation is the incorporation of two-dimensional convolutions, which is traditionally a computational burden Wernecke et al. (1977). CNN models implicitly use two-dimensional convolutions to produce feature maps therefore, the computational overheads are eliminated making the computation of MCCE loss very efficient. Furthermore, using MCCE loss a $L_1$ difference can be calculated between the reconstruction approximation $g(\mathbf{X})$ and ground truth $f(\mathbf{X})$ (CCE error).

Reconstruction error can be calculated using the apriori determined ME of the input dataset and reconstruction approximation $g(\mathbf{X})$. Monitoring the divergence between the CCE error and reconstruction error provides an indication in the degree of deviation of convolutional kernel weights and predicted class labels to the deviation of $f(\mathbf{X})$ to determine if the convolutional kernels are stuck in a global minima/maxima and thus indicating model learning has saturated.

### 3.1 ALGORITHM

The pseudo-code for MCCE loss is presented as Algorithm 1, which requires the calculation of a 1D linear interpolation output for reconstruction loss is provided in Section 3.2.

---

**Algorithm 1** Maximum Categorical Cross Entropy (MCCE)

---

1: **Input:** One-hot encoded ground truth ($y_{true}$) and CNN predicted ($y_{pred}$) class labels, apriori ME of training images in the dataset (i.e. ME($\mathbf{X}$)).
2: **Output:** Probabilistic logarithmic loss of $y_{pred}$ with the ground truth $y_{true}$.
3: **Initialize:** $\Lambda \leftarrow$ ME($\mathbf{X}$), $\mu \leftarrow$ ME($\omega$)
4: $\gamma$ = -log($\frac{e^{s_p}}{\Sigma_j^C e^{s_j}}$) {CCE loss, $s_p$ is the CNN score and $s_j$ the ground truth for the class $C$}
5: $\kappa = \Lambda - \mu$ {Convolutional Reconstruction (CR) loss}
6: $\kappa$ = Interpolation($\kappa$, (0,$\Lambda$), (0,1)) {1d linear interpolation to output $\kappa$ between 0 and 1 rather than in the range of 0 to $\Lambda$}
7: $\Delta = \gamma + \kappa$ {Maximum loss = CCE loss + CR loss}
8: return $\Delta$

---

### 3.2   1D LINEAR INTERPOLATION

A one-dimensional interpolation of the reconstruction error/loss is required as the MCCE loss is an extension of CCE loss which outputs values between 0 and 1. A linear interpolant is the straight line between the two known points given by their coordinates $(a_0, b_0)$ and $(b_1, b_1)$ Davis (1975). For any value $i$ in the interval $(a_0, a_1)$, the value of $j$ along the straight line can be calculated using the equation of slopes given in Equation 3

$$\frac{j - b_0}{i - a_0} = \frac{b_1 - b_0}{a_1 - a_0} \quad or \quad j = \frac{b_0(a_1 - i) + b_1(i - a_0)}{a_1 - a_0} \tag{3}$$

## 4   EXPERIMENTATION

Experimentation revolved around quantitatively measuring train-test divergence to determine the degree of overfitting in trained CNN models for the traditional CCE loss and our novel MCCE loss using all of the techniques discussed in Section 2. Cross-validation tests by interchanging the datasets was performed to ensure CNN models trained using MCCE loss consistently generalize better were also employed. No modifications were made to our CNN model training regime compared to the original implementation presented in He et al. (2016) apart from using different testing hardware and software frameworks (Keras with a tensorflow backend).

### 4.1   DATASETS

To determine the effect of racial bias and the efficacy of our novel MCCE loss function, we select a balanced dataset (UTKFace Zhang et al. (2017)) where each class of race/ethnicity has an equal number of images and an unbalanced dataset (colorFeret Phillips et al. (1998)) where the distribution of data across all of the classes is unequal.

The colorFERET dataset contains 11,338
semi-controlled color images of $512 \times 768$ pixel size with 13 different poses from 994 test subjects. Due to our limited computing infrastructure, the images needed to be downsampled to $96 \times 96$ pixel resolution using cubic interpolation. The original dataset contains nine classes (Asian, Asian-Southern, Asian-Middle-Eastern, Black-or-African-American,
White, Hispanic, Native-American, Other and Pacific-Islander). Due to the very limited number of test subjects and images for four of the nine classes, the dataset was reduced to five classes (Asian, Asian-Middle-Eastern, Black-or-African-American, White, Hispanic) containing a total of 11,172 images.

The original UTKFace dataset contains 23,708 in-the-wild color images of $200 \times 200$ pixel size with five ethnic classes (White, Black, Asian, Indian and Others) of all age groups. Only the OECD definition for working age population (15-64) consisting of 18,095 images are considered since the facial variations are not severe enough to cause any unexpected errors like misclassification or underfitting. The images used in our experimentation were downsampled to $96 \times 96$ pixel resolution using cubic interpolation to accommodate our limited computing infrastructure.

### 4.2   EXPERIMENTAL SETUP

All experiments presented in this paper were carried out with a single RTX 2080ti with 11GB of VRAM, generously provided by InfuseAI Limited (New-Zealand). All models were trained from scratch with the datasets randomly shuffled when reading from storage into memory and a 20% allocation of the randomly allocated dataset was reserved for testing. The training data was again randomly shuffled during model

training to mitigate any variability in the input data. This process was repeated for all the three model training instances.

Table 1: Results validating the efficacy of training models using the proposed MCCE loss function to mitigate overfitting and reduce bias

| Loss Function | Dataset | Train Acc. | Test Acc. | Train-Test $\Delta$ | Best Test Acc. |
|---|---|---|---|---|---|
| CCE | colorFERET | **97.57%** | 84.67% | 12.90% | 86.58% |
| **MCCE (Ours)** | colorFERET | 97.44% | **90.38%** | **7.05%** | **92.22%** |
| CCE | UTKFace | **94.77%** | 74.43% | 20.34% | 76.26% |
| **MCCE (Ours)** | UTKFace | 91.09% | **75.04%** | **16.05%** | **77.12%** |
| **Cross-Validation** | | | | | |
| Loss Function | Train Dataset | Test Dataset | Acc. | Best Acc. | $f$ score |
| CCE | colorFERET | UTKFace | 17.36% | 22.62% | 0.29 |
| **MCCE (Ours)** | colorFERET | UTKFace | **26.16%** | **39.91%** | **0.39** |
| CCE | UTKFace | colorFeret | 19.74% | 33.30% | 0.26 |
| **MCCE (Ours)** | UTKFace | colorFeret | **44.90%** | **63.40%** | **0.57** |

## 4.3 RESULTS

The results in Table 1 present the averaged classification performance from the three model training instances and the best test accuracy using an early stopping patience of 10, monitored on the training accuracy. Table 1 also presents the cross-validation results and $f$ scores to highlight bias in trained models on new data. MCCE trained models outperform CCE trained models in all of the measured criteria. All links to the datasets and experimental scripts used are available online and can be accessed at: *Uploaded as supplementary material*.

## 5 DISCUSSION

Analyzing the data presented in Table 1, we clearly identify the effectiveness of MCCE in model training with respect to overfitting and generalizability. Classification bias is evaluated by examining the weighted $f$ score determined using the confusion matrix. Models trained using MCCE outperform standard CCE models by almost two times. Perhaps more concerning is the fact that CCE trained models barely improve on random chance of 20% (five classes) on the balanced UTKFace cross-validation test. Although training models using MCCE might mitigate overfitting indicated by the relatively lower train-test difference, MCCE trained models still overfit to a limited degree. The tendency of relatively lower degree of overfitting holds even for cross-validation results where MCCE models achieve a greater weighted $f$ score and test accuracy.

Figures 3 and 4 illustrate the confusion matrices for both the balanced colorFERET and unbalanced UTK-Face datasets respectively. The MCCE algorithm provided enhanced classification performance for both datasets. The CCE loss was especially vulnerable to unbalanced datasets, suggesting implicit biases with respect to the training set significantly corrupts the final results. Similar patterns of implicit biases can be observed for the MCCE loss however not as pronounced relative to CCE trained models demonstrating the improved resilience of the MCCE algorithm.

MCCE trained models had a relatively higher loss of 1.36 and 1.5 compared to 0.35 and 0.38 for the colorFERET and UTKFace datasets respectively, suggesting a greater degree of $L_2$ kernel regularization is being employed by the MCCE algorithm. The higher loss indicates the MCCE trained models have not fully

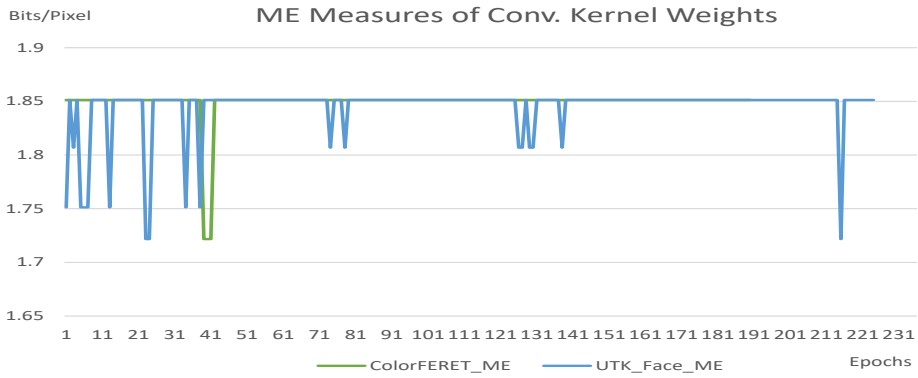

Figure 2: ME measures of the convolutional kernel weights during model training using $r = 1$ for color-FERET and UTKFace datasets

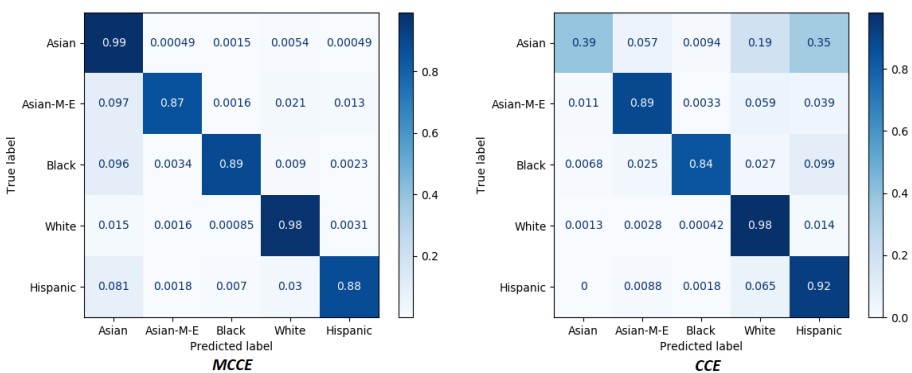

Figure 3: Confusion matrix for the five classes of the balanced colorFERET dataset; Asian-M-E: Asian Middle-Eastern

converged and greater improvements can be achieved with manual HP fine tuning. Furthermore, MCCE trained models generally converge faster taking an average of 176 and 204 epochs compared to 199 and 268 epochs for the colorFERET and UTKFace datasets respectively. Faster convergence along with higher losses suggests an enhanced learning capacity of the MCCE models, which can be improved with exploration of additional techniques to improve convergence such as learning rates.

Figure 2 illustrates the convolutional kernel weight ME measures during model training for both datasets, which does not reach the 2.09 and 2.25 bits per pixel measures for colorFERET and UTKFace datasets respectively. Examining Figure 2 highlights the MCCE model training process where large divergences from the maximum are quickly penalized and weight adjustments during the back-propagation step correct these divergences quickly. In Figure 5, we visualize the loss curves for CCE and MCCE loss functions during model training. MCCE and CCE loss functions generally follow the same curve relative to each other but differ in their final convergence; a similar pattern can be observed in Figure **??** for model training.

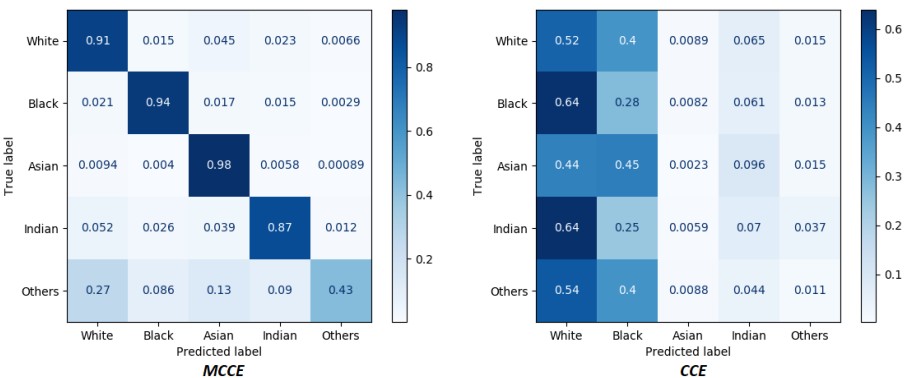

Figure 4: Confusion matrix for the five classes of the unbalanced UTKFace dataset

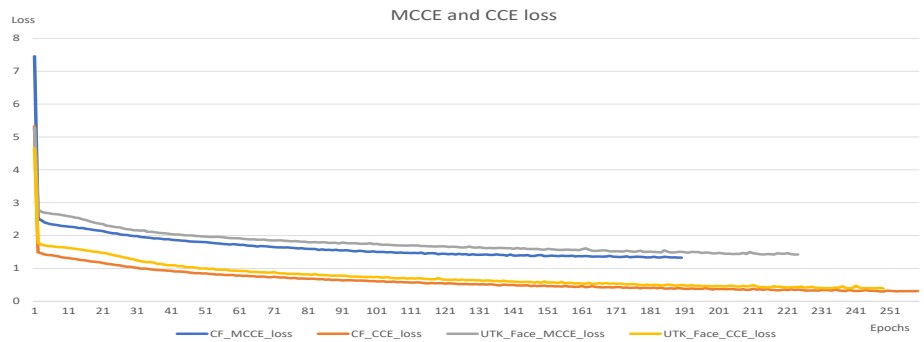

Figure 5: Loss curves for MCCE and CCE loss functions for colorFERET and UTKFace datasets

## 6  CONCLUSION AND FUTURE WORK

In this paper, we proposed a novel extension to the commonly used Categorical Cross Entropy (CCE) loss function known as Maximum Categorical Cross Entropy (MCCE). While CCE evaluates the probability distributions of the CNN predicted and ground truth class labels, MCCE extends this evaluation to include the entropic distribution of convolutional kernel weights during model training. MCCE provides a robust noise-averse method of calculating model loss since partial knowledge of the entropic distribution of the input data is determined by a priori and large divergences from the maximum are penalized during model training.

MCCE loss takes into account the label loss and convolution kernel weight distribution loss or reconstruction loss penalizing model training if either of these distributions greatly diverge from optimal. MCCE loss has been empirically validated to minimize overfitting by 5.85% and 4.3% using a ResNet architecture on the ColorFeret and UTKFace datasets respectively. Furthermore, MCCE has shown to improve generalizability of trained models in cross-validation testing by 8.8% using the trained colorFeret models on UTKFace and 25.16% using the UTKFace trained models on colorFERET. The knowledge of entropic distribution of convolutional kernel weights during model training can be used to determine the state of convergence of the model. This state determination can be used to adjust other model parameters like learning rates which are reserved as our future work.

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
