# OpenReview forum: "Maximum Categorical Cross Entropy (MCCE): A noise-robust alternative loss function to mitigate racial bias in Convolutional Neural Networks (CNNs) by reducing overfitting"
_ICLR.cc/2021/Conference — Reject_

### Official Review · AnonReviewer3 · 2020-10-27
**review for "Maximum Categorical Cross Entropy (MCCE): A noise-robust alternative loss function to mitigate racial bias in Convolutional Neural Networks (CNNs) by reducing overfitting""**

**Rating:** 3
**Confidence:** 4

**Review:**

In this paper, the author studies the bias problem in race classification task with face data. Specifically, it first analyses the influence of kernel regularization and batch normalization to categorical cross-entropy loss and proposes a maximum categorical cross-entropy loss. Experiments on two face datasets colorFERET and UTKFace demonstrate the effectiveness of the proposed method.

From the ethical aspect, the topic of this paper is important and interesting. But it seems the proposed loss is not specially designed for the racial bias problem, please consider evaluate the proposed loss on general image classification tasks. There have some losses for imbalanced training can be used in this topic (e.g. Focal loss, GHM-C loss). In this paper, the author only compares their approach with the traditional CCE loss which is not convincing. And also, to show the proposed approach can mitigate the bias problem in the race classification task, the author should show the accuracy for different races rather than an averaged accuracy. Overall, I think this paper’s topic is important but the approach seems not make sense and less relevant to the racial bias problem.

Pros.
1.	The bias problem that this paper studied is an important problem for image classification, especially for race classification.
2.	The results in the experiment section could partially demonstrate the effectiveness of the proposed MCCE loss.

Cons.
1.	The writing of this paper is bad and hard to follow. The author uses several subsections (Section 2.1~2.3, 3.2) to introduce the cross-entropy loss, kernel regularization, batch normalization, and linear interpolation which are redundant.
2.	Algorithm 1 is not aligned with the paper. The variable \mu is not used in the algorithm but seems to be very important to the method (see Section 3).
3.	Accuracy, the key evaluation metric in the experiment part, cannot fully demonstrate the effectiveness of the proposed method. Please consider adding confusion matrix or per-class accuracy.
4.	The discussion section (Section 5) seems not clear. The figures in that section (ME measures, loss curve, training accuracy) cannot support the conclusions.
5.	There are some formatting problems in the paper (Page 7 Line 1 & 5). The figures in the paper look like screenshots from Excel which are not very clear. Please consider inserting the figures in a vectorized format.

Overall,  I think this paper is far more below the ICLR acceptance bar.

---

### Official Review · AnonReviewer1 · 2020-10-27
**This paper proposes an extension to the traditional Categorical Cross Entropy Loss known as the Maximum Categorical Cross Entropy (MCCE) Loss.**

**Rating:** 5
**Confidence:** 4

**Review:**

Pros:

1.The authors propose an extension of the CE loss to reduce classification bias that occurs in present methods and datasets. They calculate Maximum Entropy (ME) for images on the entire training dataset and then calculate the reconstruction loss between this and the ME for convolutional kernels during training. Their experiments results show that minimizing this reconstruction loss along with CE speeds up convergence.
2.The paper is thoroughly written with minor typos and is easy to follow.


Cons:

1.How does minimizing Maximum Entropy in the form of reconstruction error help to improve the weights learned by the model more suited for unbiased performance ? As shown it might help in faster learning but it’s not very clear how and why it learns good feature maps?
2.How is the ME entropy calculated? It is good to briefly discuss the method/formula to calculate that.
3.In Algorithm 1, is there an error in line 5, it should be mu instead of gamma? The 1D interpolation seems like a good way to normalize, but is it the only way or the best way?
Also discuss some experimental results with per-class accuracy/precision/recall in case of unbalanced datasets.
4.It seems like this loss acta as a regularizer on the CE loss only training, but does that also help as a prior knowledge or information to learn better information as discussed in paper several times. Some evaluation on this either comparing feature activations or particularly which category improves more compared to CE might give a better intuition.
5.Please discuss some related work  or compare against as baselines with papers also trying to reduce classification bias.

---

### Official Review · AnonReviewer2 · 2020-10-27
**The current write up is unclear and needs improvement**

**Rating:** 4
**Confidence:** 5

**Review:**

Summary: the paper proposes a new loss function, called MCCE to reduce the effect of overfitting to noisy examples. This involves calculating the Maximum Entropy (ME) of the input images as well as the filters (?). Experiments are conducted on standard datasets to validate the claims.

Review: I find the current state of the paper very confusing and unclear. Specifically, it is unclear what the method is trying to optimize (other than adding some form of regularization term based on entropy). The only technical development of the algorithm is given in Alg. 1 and no justification is provided for the design choices (such as: how is mu = ME(w) used in the algorithm? What does convolutional reconstruction loss amount to? What is the purpose of the interpolation? ...). The general discussion up to Section 2 can be shortened significantly and devoted to the development of the method. Overall, the paper is poorly written on the technical side.

Additionally, it is not clear why the authors attribute the bias in the predictions to noisy examples. For instance, a poorly trained model or a model which overfits to certain examples can produce biased predictions. A number of recent work also aim to reduce the effect of overfitting to noisy examples. For instance, (Amid et al. 2019a) generalizes the GCE loss (Zhang and Sabuncu 2018) by introducing two temperatures t1 and t2 which recovers GCE when t1 = q and t2 = 1. A more recent work, called the bi-tempered loss (Amid et al. 2019b) extends these methods by introducing a proper (unbiased) generalization of the CE loss and is shown to be extremely effective in reducing the effect of noisy examples. Also, (Yang and Guo 2020) proposes peer-loss (which can be combined with CE, bi-tempered, etc. loss) for handling noise. Please consider referencing/comparing to these SOTA methods.

Additional references:

(Amid et al. 2019a) Amid et al. "Two-temperature logistic regression based on the Tsallis divergence." In The 22nd International Conference on Artificial Intelligence and Statistics, 2019.

(Amid et al. 2019b) Amid et al. "Robust bi-tempered logistic loss based on Bregman divergences." In Advances in Neural Information Processing Systems, 2019.

(Yang and Guo 2020) Yang and Guo. "Peer Loss Functions: Learning from Noisy Labels without Knowing Noise Rates." In International Conference on Machine Learning, 2020.

---

### Official Review · AnonReviewer4 · 2020-10-29
**an extension to the categorical cross entropy to reduce overfitting**

**Rating:** 5
**Confidence:** 3

**Review:**

The paper proposes a new extension to the categorical cross-entropy using maximum entropy (MCCE) loss function to reduce model overfitting. The goal is to stabilize the training with respect to overfitting and generalizability.
Strengths:
+The proposed method is simple and elegant. It is theoretically well-founded and easily implemented.
+The paper provides good initial results, and the experiments are conducted on the various dataset: colorFERET and UTKFace.

Weakness:
+ The proposed method is not novel and just a combining of maximum entropy with cross-entropy.
+ The authors claimed to upload the supplementary material, but it's missing.
+ The authors should describe the detailed CNN models implemented with MCCE. And should report the computation cost for each experiment.
+  More generalization analysis would be beneficial.

---

### Review · Ethics_Committee · 2021-01-06

**Decision:**

No judgement (proceed with normal process)

**Ethics Review:**

This paper was flagged for evaluation by the ethics board based on the following:
1.	the motivation of the paper is to reduce model overfitting and racial bias towards one category. However, there is no further discussion about any "ethical, societal and practical concerns when dealing with facial datasets, especially for the task of race or gender classification".

The primary concern regarding race bias in computer vision applications has been the question of whether systems that perform face recognition (recognizing a known person from a novel image of that person) or face attribute prediction (e.g., predicting gender or age) perform worse on some racial groups such as people with darker skin. This paper does not address that question. Instead, it trains a classifier to predict the race of the person. While this may tell us something about how classifiers can confuse different racial groups, it doesn’t tell us anything about how race biases performance on other tasks.

To study the efficacy of their MCCE loss function, the authors selected two facial image datasets consisting of various racial/ethnicity classes (e.g. White, Hispanic, Black, Asian, etc.) but with either a balanced or unbalanced number of images across classes (in one dataset authors have aggregated classes on the basis of already biased datasets due to the “very limited images” for particular classes). They then illustrated the impact of these datasets on classification performance. The authors claim that “implicit biases with respect to the training set significantly corrupts the final results”. However, there is no reason to blame “implicit biases”. The observed results could simply be due to class imbalance, which is completely explicit, not implicit. It is possible that the MCCE loss function would reduce racial bias, as it appears to do a better job of normalizing image features. But the present paper cannot distinguish
this from the class imbalance hypothesis. Furthermore, the fact that CCE does pretty well when trained on a balanced dataset provides additional evidence that class imbalance is the primary culprit. To separate these two potential explanations, the authors need to separately vary race and class imbalance. For that purpose, it would be best to focus on a different primary prediction task (e.g., gender or age classification) and then compare class imbalance to changes in racial composition (e.g., all images from a single race vs. from multiple races).

The authors state that UTKFace dataset is imbalanced, but they do not provide statistics on the imbalance (nor does the original paper that created the dataset). Based on the confusion matrix results, I’m guessing that White faces form a huge majority class in this dataset. Every other class is confused with the White class (by the CCE loss).

The authors are claiming that they are mitigating racial bias, but because they do not demonstrate a task that exhibits racial bias (e.g., face recognition or face gender or age prediction), this claim is not supported. We recommend that they remove this claim from the paper. Instead, their results would appear to support claims of (a) improved accuracy in the presence of class imbalance and (b) improved domain generalization. Both of these claims are plausible results of their improved loss function.

If the authors do want to make a claim of reducing racial bias, then they should study a task such as face recognition or face attribute detection and measure how performance on those tasks varies with race (e.g., along the lines of the famous Gender Shades paper, Buolamwini, J., & Gebru, T. (2018). Gender Shades: Intersectional Accuracy Disparities in Commercial Gender Classification. Proceedings of Machine Learning Research, 81, 1–15.). Furthermore, as in the Buolamwini & Gebru paper, they should use an objective measure of skin darkness/lightness rather than relying on race labels whose meaning can vary greatly.

The questions being studied in this paper are important for improving the fairness of computer vision systems and do not raise ethical concerns. Rather, we find that the authors have not demonstrated that they are making progress on those issues.

Decision: No Concern (i.e. no judgement as indicated)

---

### Decision · Program_Chairs · 2021-01-07
**Final Decision**

**Decision:**

Reject

**Comment:**

The paper introduces a new loss, Maximum Categorical Cross-Entropy, which combines the usual cross-entropy loss with a maximum entropy regularisation term on the convolutional kernels, and is evaluated on image classification. The authors have trained a face classification algorithm on two datasets: UTKFace (https://susanqq.github.io/UTKFace/) and NIST colorFERET (https://www.nist.gov/itl/products-and-services/color-feret-database). The labels consisted in, respectively: White, Black, Indian, Asian, Others (over 18k images) and Asian, Asian-Middle-Easter, Black-of-African-American, White, Hispanic (over 11k images) (see section 4.1 of the paper).

From the meta-reviewer's perspective:
As stated in the title, abstract and in paragraph 3 of section 1, the motivation of the paper is to reduce model overfitting and racial bias towards one category. However, there is no further discussion about any "ethical, societal and practical concerns when dealing with facial datasets, especially for the task of race or gender classification". It seems to me that a paper that implements a "race classification" algorithm should at least devote a substantially long part of the discussion on the validity of such a task and of such a labelling process, as well as question the motivations and potential misuses. Who labeled these faces and based on what visual characteristics? Were the subjects of the photographs consenting and did they self-declare their ethnicities? Are the authors simply reproducing discredited phrenology assumptions about ethnicities and about "race", which is increasingly defined as a mere social construct? Given that there is nothing specific to face classification in the loss function, I wonder why did the authors decide to focus on ethnicity features? What exactly could a visual ethnicity classifier be used for? Given the sheer amount of questions raised by the paper, we have submitted it for review by the Ethics Board.

Summary of the reviews:
Reviewers gave scores scores 3, 4, 5, 5 (without rebuttal from the authors), raising concerns about the novelty and contribution of the method (as it is simply combining maximum entropy with cross-entropy), clarity of the explanation of the method, missing related work and baselines and evaluation metrics.

Based on the low scores, unfavourable reviews and an ongoing Ethics Board investigation, I recommend for this paper to be rejected.



While this paper is likely to be rejected (, I believe that these concerns should be raised and potentially reviewed by the Ethics Board (unless this is an obvious rejection). Thank you in advance for your time.